# Diffusion Models for Open-Vocabulary Segmentation

Laurynas Karazija    Iro Laina    Andrea Vedaldi    Christian Rupprecht

Visual Geometry Group, University of Oxford

https://www.robots.ox.ac.uk/~vgg/research/ovdiff/

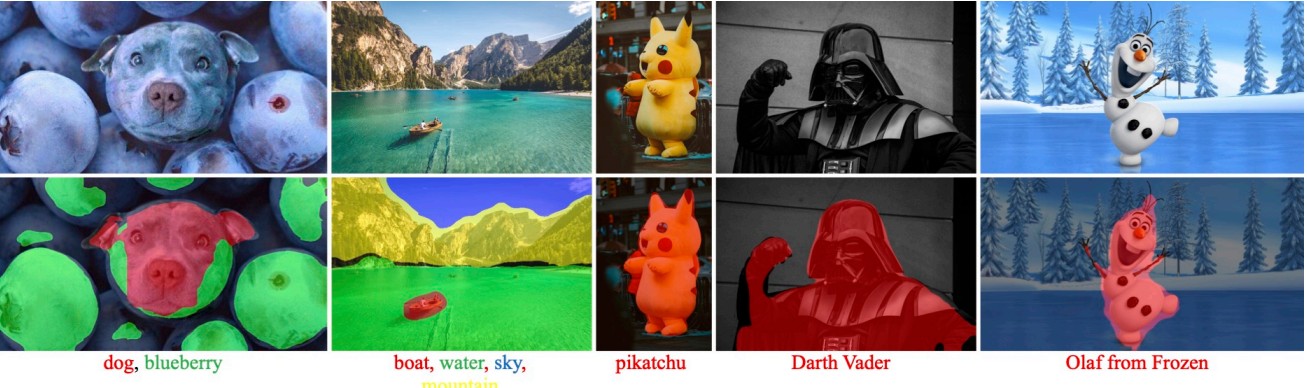

dog, blueberry      boat, water, sky, mountain      pikatchu      Darth Vader      Olaf from Frozen

Figure 1. OVDiff is an open-vocabulary segmentation method that, given an image and a free-form set of class names, can segment any user-defined classes. It is fully automatic and does not require any further training.

## Abstract

*Open-vocabulary segmentation is the task of segmenting anything that can be named in an image. Recently, large-scale vision-language modelling has led to significant advances in open-vocabulary segmentation, but at the cost of gargantuan and increasing training and annotation efforts. Hence, we ask if it is possible to use* existing *foundation models to synthesise on-demand efficient segmentation algorithms for specific class sets, making them applicable in an open-vocabulary setting without the need to collect further data, annotations or perform training. To that end, we present OVDiff, a novel method that leverages generative text-to-image diffusion models for unsupervised open-vocabulary segmentation. OVDiff synthesises support image sets for arbitrary textual categories, creating for each a set of prototypes representative of both the category and its surrounding context (background). It relies solely on pre-trained components and outputs the synthesised segmenter directly, without training. Our approach shows strong performance on a range of benchmarks, obtaining a lead of more than 5% over prior work on PASCAL VOC.*

## 1. Introduction

Open-vocabulary semantic segmentation is the task of segmenting images into regions matching several free-form textual categories. As the field of Computer Vision moves towards large-scale general-purpose models, open-vocabulary "foundation" models have similarly emerged. Yet, the development of ones suitable for dense localisation tasks such as semantic segmentation incurs both enormous training costs and requires expensive mask annotations. Instead, we show that the open-vocabulary segmentation task can be effectively tackled starting from a set of frozen foundation models, without requiring additional data or even fine-tuning.

In order to do so, we introduce OVDiff, a method that turns existing foundation models into a "factory" of image segmenters, *i.e.*, using foundation models to synthesise on-demand a segmenter for any new concepts specified in natural language. Thus, OVDiff can be used for open-vocabulary segmentation, where it achieves state-of-the-art results in standard benchmarks. Moreover, once synthesised, the segmenters can be efficiently applied to any number of images and easily extended to new categories.

Specifically, segmenting an image using OVDiff can be done in three steps: *generation*, *representation*, and *matching*. Given a textual prompt, OVDiff uses an off-the-shelf text-to-image generator like StableDiffusion [50] to *generate*

a support set of images. In the representation step, we use a feature extractor (that can be the same network as in the generation step) to extract feature prototypes that represent the textual category. Finally, we use simple nearest-neighbour *matching* scheme to segment the target image using the prototypes computed in the previous step.

This approach differs from prior work that largely approaches the problem in either of two ways. Starting from multi-modal representations (*e.g.*, CLIP [46]) to bridge vision and language, the first way relies on labelled data to fine-tune image-level representations for the segmentation task. Hence, in line with the zero-shot setting [6], these methods require costly dense annotations for some known categories while also extending the segmentation to unseen categories by incorporating language.

The second category of prior work [9, 37, 43, 49, 70, 71] observes that large-scale vision-language models such as CLIP have a limited understanding of the positioning of objects within an image and extend these models with additional grouping mechanisms for better localisation using only image-level captions, but no mask supervision. This, however, requires expensive additional contrastive training at scale. Despite yielding promising results, there are some additional pitfalls to this approach. Firstly, as the text might not exhaustively describe all entities in the image or might mention elements that are not depicted, the training signal can be noisy. Secondly, similar captions may be used to describe a wide range of visual appearances, or a similar concept might be described differently, sometimes even depending on the other context present. There is ambiguity and a difference in detail between visual and textual data. Lastly, most methods resort to heuristics to segment the background (*i.e.*, leave some pixels unlabelled), as it often cannot be described as a textual category. The usual approach is to threshold the similarities to all categories. Finding an appropriate threshold, however, can be challenging and may vary depending on the image, often resulting in imprecise object boundaries. Effectively handling the background remains an open issue.

Our three-step approach departs substantially from both of these schemes. We show that large-scale text-to-image generative models, such as StableDiffusion [50], can help bridge the vision-and-language gap without the need for annotations or costly training. Furthermore, diffusion models also produce latent spaces that are semantically meaningful and well-localised. This solves a second problem: multi-modal embeddings are difficult to learn and often suffer from ambiguities and differences in detail between modalities. Instead, our approach can use unimodal features for open-vocabulary segmentation, which offers several advantages. Firstly, as text-to-image generators encode a distribution of possible images, this offers a means to deal with intra-class variation and captures the ambiguity in textual descriptions.

Secondly, the generative image models encode not only the visual appearance of objects but also provide contextual priors, which we use for direct background segmentation.

This work presents a simple framework that achieves state-of-the-art performance across open-vocabulary segmentation benchmarks. It combines several off-the-shelf pre-trained networks into a segmenter "factory" that segments images into arbitrary textual categories in three simple steps. OVDiff requires no additional data, mask supervision, nor fine-tuning. To summarise, we make the following core contributions: (1) We introduce a method to use pre-trained diffusion models for the task of open-vocabulary segmentation, that requires no additional data, mask supervision, or fine-tuning. (2) We propose a principled way to handle backgrounds by forming prototypes from contextual priors built into text-to-image generative models. (3) A set of additional techniques for further improving performance, such as multiple prototypes, category filtering and "stuff" filtering.

## 2. Related work

**Zero-shot open-vocabulary segmentation.** Open-vocabulary semantic segmentation is a relatively new problem and is typically approached in two ways. The first line of work poses the problem as "zero-shot", *i.e.*, segmenting unseen classes after training on a set of observed classes with dense annotations. Early approaches [6, 11, 20, 31] explore generative networks to sample features using conditional language embeddings for classes. In [30, 69] image encoders are trained to output dense features that can be correlated with word2vec [41] and CLIP [46] text embeddings. Follow-up works [15, 19, 33, 73] approach the problem in two steps, predicting class-agnostic masks and aligning the embeddings of masks with language. IFSeg [74] generates synthetic feature maps by pasting CLIP text embeddings into a known spatial configuration to use as additional supervision. Different from our approach, all these works rely on mask supervision for a set of known classes.

The second line of work eliminates the need for mask annotations and instead aims to align image regions with language using only image-text pairs. This is largely enabled by recent advancements in large-scale vision-language models [46]. Some methods introduce internal grouping mechanisms such as hierarchical grouping [49, 70], slot-attention [71], or cross-attention to learn cluster centroids [35, 37]. Assignment to language queries is performed at group level. Another line of work [9, 43, 48, 79] aims to learn dense features that are better localised when correlated with language embeddings at pixel level. With the exception of [48, 68, 79], thresholding is often required to determine the background during inference. Alternatively, a curated list of background prompts can be used [48].

Our method falls into the second category. However,

in contrast to prior work, we leverage a generative model to translate language queries to pre-trained image feature extractors without further training. We also segment the background directly, without relying on thresholding or curated list of background prompts. A closely related approach to ours is ReCO [56], where CLIP is used for image retrieval compiling a set of exemplar images from ImageNet for a given language query, which is then used for co-segmentation. In our method, the shortcoming of an image database is addressed by synthesising data on-demand. Furthermore, instead of co-segmentation, we leverage the cross-attention of the generator to extract objects. Instead of similarity of support images, we use diverse samples and both foreground and contextual backgrounds.

**Diffusion models.** Diffusion models [26, 59, 60] are a class of generative methods that have seen tremendous success in text-to-image systems such as DALL-E [47], Imagen [52], and Stable Diffusion [50], trained on Internet-scale data such as LAION-5B [54]. The step-wise generative process and the language conditioning make pre-trained diffusion models attractive also for discriminative tasks. They have been recently used in few-shot classification [77], few-shot segmentation [2] and panoptic segmentation [72], and to generate pairs of images and segmentation masks [32]. However, these methods rely on dense manual annotations to associate diffusion features with the desired output.

Annotation-free discriminative approaches such as [13, 29] use pre-trained diffusion models as zero-shot classifiers. DiffuMask [67] uses prompt engineering to synthesise a dataset of "known" and "unseen" categories and trains a closed-set segmenter with masks obtained from the cross-attention maps of the diffusion model. DiffusionSeg [38] uses DDIM inversion [60] to obtain feature maps and attention masks of object-centric images to perform unsupervised object discovery, but relies on ImageNet labels and is not open-vocabulary. Our approach also leverages the rich semantic information present in diffusion models for segmentation; unlike these methods, however, it is open-set and does not require further training.

**Unsupervised segmentation.** Our work is also related to unsupervised segmentation approaches. While early works relied on hand-crafted priors [12, 44, 66, 75, 76] later approaches leverage feature extractors such as DINO [8] and perform further analysis of these methods [21, 39, 55, 57, 58, 63–65]. Some approaches make use of generative methods, usually GANs, to separate images in foreground and background layers [3–5, 10] or analyse latent structure to induce known foreground-background changes [40, 62] to synthesise a training dataset with labels. Largely focused on unsupervised saliency prediction, these methods are class-agnostic and do not incorporate language.

## 3. Method

We present OVDiff, a method for open-vocabulary segmentation, *i.e.*, semantic segmentation of any category described in natural language. We achieve this goal in three steps: (1) we leverage text-to-image generative models to *generate* a set of images representative of the described category, (2) use these to ground *representations* from off-the-shelf pretrained feature extractors, and (3) *match* these against input image features to perform segmentation.

### 3.1. OVDiff: Diffusion-based open-vocabulary segmentation

Our goal is to devise an algorithm which, given a new vocabulary of categories $c_i \in \mathcal{C}$ formulated as natural language queries, can segment any image against it. Let $I \in \mathbb{R}^{H \times W \times 3}$ be an image to be segmented. Let $\Phi_v : \mathbb{R}^{H \times W \times 3} \to \mathbb{R}^{H'W' \times D}$ be an off-the-shelf visual feature extractor and $\Phi_t : \mathbb{R}^{d_t} \to \mathbb{R}^D$ a text encoder. Assuming that image and text encoders are aligned, one can achieve segmentation by simply computing a similarity function, for example, the cosine similarity $s(\Phi_v(I), \Phi_t(c_i))$, with $s(x,y) = \frac{x^T y}{\|x\|\|y\|}$, between the encoded image $\Phi_v(I)$ and an encoding of a class label $c_i$. To meaningfully compare different modalities, image and text features must lie in a shared representation space, which is typically learned by jointly training $\Phi_v$ and $\Phi_t$ using image-text or image-label pairs [46].

We propose two modifications to this approach. First, we observe that it is better to compare representations of the *same* modality than across vision and language modalities. We thus replace $\Phi_t(c_i)$ with a $D$-dimensional *visual* representation $\bar{P}$ of class $c_i$, which we refer to as a *prototype*. In this case, the same feature extractor can be used for both prototypes and target images; thus, their comparison becomes straightforward and does not necessitate further training. Second, we propose utilising *multiple* prototypes per category instead of a single class embedding. This enables us to accommodate intra-class variations in appearance, and, as we explain later, it also allows us to exploit contextual priors, which in turn help to segment the background.

Our approach, thus, proceeds in three steps: (1) a set of support images is sampled based on vocabulary $\mathcal{C}$, (2) a set of prototypes $\mathcal{P}$ is calculated, and (3) a set of images $\{I_1, I_2 \dots\}$ is segmented against these prototypes. We observe that in practical applications, whole image collections are processed using the same vocabulary, as altering the set of target classes for individual images in an informed way would already require some knowledge of their contents. Steps (1) and (2) are, thus, performed very infrequently, and their cost is heavily amortised. Next, we detail each step.

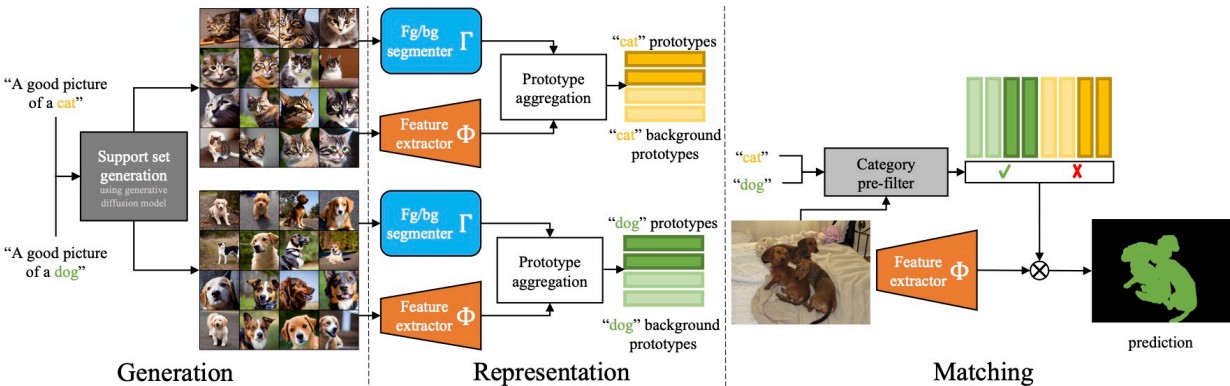

Figure 2. OVDiff overview. Prototype sampling: text queries are used to sample a set of support images which are further processed by a feature extractor and a segmenter forming positive and negative (background) prototypes. Segmentation: image features are compared against prototypes. The CLIP filter removes irrelevant prototypes based on global image contents.

## 3.2. Support set generation

To construct a set of prototypes, the first step of our approach is to sample a support set of images representative of each category $c_i$. This can be accomplished by leveraging pre-trained text-conditional generative models. Sampling images from a generative model, as opposed to a curated dataset of real images, aligns well with the goals of open-vocabulary segmentation as it enables the construction of prototypes for *any* user-specified category or description, even those for which a manually labelled set may not be readily available (*e.g.*, $c_i$ = "donut with chocolate glaze").

Specifically, for each query $c_i$, we define a prompt "A good picture of a $\langle c_i \rangle$" and generate a small batch of $N$ support images $\mathcal{S} = \{S_1, S_2, \ldots, S_N \mid S_n \in \mathbb{R}^{hw \times 3}\}$ of height $h$ and width $w$ using Stable Diffusion [50].

## 3.3. Representing categories

Naïvely, prototypes $\bar{P}_{c_i}$ could be constructed by averaging all features across all images for class $c_i$. This is unlikely to result in good prototypes because not all pixels in the sampled images correspond to the class specified by $c_i$. Instead, we propose to extract the class prototypes as follows.

**Class prototypes.** Our approach generates two sets of prototypes, positive and negative, for each class. Positive prototypes are extracted from image regions that are associated with $\langle c_i \rangle$, while negative prototypes represent "background" regions. Thus, to obtain prototypes, the first step is segmenting the sampled images into foreground and background. To identify regions most associated with $c_i$, we use the fact that the layout of a generated image is largely dependent on the cross-attention maps of the diffusion model [24], *i.e.*, pixels attend more strongly to words that describe them. For a given word or description (in our case $c_i$), one can generate a set of attribution maps $\mathcal{A} = \{A_1, A_2, \ldots, A_N \mid A_n \in \mathbb{R}^{hw}\}$, corresponding to the support set $\mathcal{S}$, by summing the cross-attention maps across all layers, heads, and denoising steps

of the network [61].

Yet, thresholding these attribution maps may not be optimal for segmenting foreground/background, as they are often coarse or incomplete, and sometimes only parts of objects receive high activation. To improve segmentation quality, we propose to optionally leverage an unsupervised instance segmentation method $\Gamma$. Unsupervised segmenters are not vocabulary-aware and may produce multiple binary object proposals. We denote these as $\mathcal{M}_n = \{M_{nr} \mid M_{nr} \in \{0,1\}^{hw}\}$, where $n$ indexes the support images and $r$ indexes the object masks (including a mask for the background). We thus construct a promptable extension of $\Gamma$ segmenter to select appropriate proposals for foreground and background: for each image, we select from $\mathcal{M}_n$ the mask with the highest (lowest) average attribution as the foreground (background):

$$M_n^{\text{fg}} = \arg\max_{M \in \mathcal{M}_n} \frac{M^\top A_n}{M^\top M}, \quad M_n^{\text{bg}} = \arg\min_{M \in \mathcal{M}_n} \frac{M^\top A_n}{M^\top M}. \tag{1}$$

**Prototype aggregation.** We can compute prototypes $P_n^{\text{g}}$ for foreground and background regions ($\text{g} \in \{\text{fg}, \text{bg}\}$) as

$$P_n^{\text{g}} = \frac{(\hat{M}_n^{\text{g}})^\top \Phi_v(S_n)}{m_n^{\text{g}}} \in \mathbb{R}^D, \tag{2}$$

where $\hat{M}_n^{\text{g}}$ denotes a resized version of $M_n^{\text{g}}$ that matches the spatial dimensions of $\Phi_v(S_n)$, and $m_n^{\text{g}} = (\hat{M}_n^{\text{g}})^\top \hat{M}_n^{\text{g}}$ counts the number of pixels within each mask. In other words, prototypes are obtained by means of an off-the-shelf pretrained feature extractor and computed as the average feature within each mask.

We refer to these as *instance* prototypes because they are computed from each image individually, and each image in the support set can be viewed as an instance of class $c_i$.

In addition to instance prototypes, we found it helpful to also compute *class-level* prototypes $\bar{P}^{\text{g}}$ by averaging the

instance prototypes weighted by their mask sizes as $\bar{P}^g = \sum_{n=1}^{N} m_n^g P_n^g / \sum_{n=1}^{N} m_n^g$.

Finally, we propose to augment the set of class and instance prototypes using $K$-Means clustering of the masked features to obtain *part-level* prototypes. We perform spatial clustering separately on foreground and background regions and take each cluster centroid as a prototype $P_k^g$ with $1 \leq k \leq K$. The intuition behind this is to enable segmentation at the level of parts, support greater intra-class variability, and a wider range of feature extractors that might not be scale invariant.

We consider the union of all these feature prototypes:

$$\mathcal{P}^g = \bar{P}^g \cup \{P_n^g \mid 1 \leq n \leq N\} \cup \{P_k^g \mid 1 \leq k \leq K\} \quad (3)$$

for $g \in \{\mathrm{fg}, \mathrm{bg}\}$, and associate them with a single category.

We note that this process is repeated for each $c_i \in \mathcal{C}$ and we hereby refer to $\mathcal{P}^{\mathrm{fg}}$ (and $\mathcal{P}^{\mathrm{bg}}$) as $\mathcal{P}_{c_i}^{\mathrm{fg}}$ ($\mathcal{P}_{c_i}^{\mathrm{bg}}$), *i.e.*, as the foreground (background) prototypes of class $c_i$.

Since $\mathcal{P}_{c_i}^{\mathrm{fg}}$ ($\mathcal{P}_{c_i}^{\mathrm{bg}}$) depend only on class $c_i$, they can be precomputed, and the set of classes can be dynamically expanded without the need to adapt existing prototypes.

### 3.4. Segmentation via prototype matching

To perform segmentation of any target image $I$ given a vocabulary $\mathcal{C}$, we first extract image features using the same visual encoder $\Phi_v$ used for the prototypes. The vocabulary is expanded with an additional background class $\hat{\mathcal{C}} = \{c_{\mathrm{bg}}\} \cup \mathcal{C}$, for which the positive (*foreground*) prototype is the union of all *background* prototypes in the vocabulary: $\mathcal{P}_{c_{\mathrm{bg}}}^{\mathrm{fg}} = \bigcup_{c_i \in \mathcal{C}} \mathcal{P}_{c_i}^{\mathrm{bg}}$. Then, a segmentation map can simply be obtained by matching dense image features to prototypes using cosine similarity. A class with the highest similarity in its prototype set is chosen:

$$M = \arg\max_{c \in \hat{\mathcal{C}}} \max_{P \in \mathcal{P}_c^{\mathrm{fg}}} s(\Phi_v(I), P). \quad (4)$$

**Category pre-filtering.** To limit the impact of spurious correlations that might exist in the feature space of the visual encoder, we introduce a pre-filtering process for the target vocabulary given image $I$. Specifically, we leverage CLIP [46] as a strong open-vocabulary classifier but propose to apply it in a multi-label fashion to constrain the segmentation to the subset of categories $\mathcal{C}' \subseteq \mathcal{C}$ that appear in the target image. First, we encode the target image and each category using CLIP. Any categories that do not score higher than $1/|\mathcal{C}|$ are removed from consideration, that is we keep the subset $\{P_{c'}^{\mathrm{g}} \mid c' \in \mathcal{C}'\}$, $\mathrm{g} \in \{\mathrm{fg}, \mathrm{bg}\}$. If more than $\eta$ categories are present, then the top-$\eta$ are selected. We then form "multi-label" prompts as "$\langle c_a \rangle$ and $\langle c_b \rangle$ and ..." where the categories are selected among the top scoring ones taking into account all $2^\eta$ combinations. The best-scoring multi-label prompt determines the final list of categories to be used in Equation (4).

Table 1. Open-vocabulary segmentation. Comparison of our approach, OVDiff, to the state of the art (under the mIoU metric). Our results are an average of 5 seeds $\pm\sigma$. *results from [9].

| Method | Support Set | Further Training | VOC | Context | Object |
|---|---|---|---|---|---|
| ReCo* [56] | Real | ✗ | 25.1 | 19.9 | 15.7 |
| ViL-Seg [35] | ✗ | ✓ | 37.3 | 18.9 | - |
| MaskCLIP* [79] | ✗ | ✗ | 38.8 | 23.6 | 20.6 |
| TCL [9] | ✗ | ✓ | 51.2 | 24.3 | 30.4 |
| CLIPpy [48] | ✗ | ✓ | 52.2 | - | 32.0 |
| GroupViT [70] | ✗ | ✓ | 52.3 | 22.4 | - |
| ViewCo [49] | ✗ | ✓ | 52.4 | 23.0 | 23.5 |
| SegCLIP [37] | ✗ | ✓ | 52.6 | 24.7 | 26.5 |
| OVSegmentor [71] | ✗ | ✓ | 53.8 | 20.4 | 25.1 |
| CLIP-DIY [68] | ✗ | ✗ | 59.9 | – | 31.0 |
| **OVDiff** (-CutLER) | Synth. | ✗ | 62.8 | 28.6 | 34.9 |
| **OVDiff** | Synth. | ✗ | **66.3 ± 0.2** | **29.7 ± 0.3** | **34.6 ± 0.3** |
| TCL [9] (+PAMR) | ✗ | ✓ | 55.0 | 30.4 | 31.6 |
| **OVDiff** (+PAMR) | Synth. | ✗ | **68.4 ± 0.2** | **31.2 ± 0.4** | **36.2 ± 0.4** |

Table 2. Segmentation performance of OVDiff based on different feature extractors.

| Feature Extractor | MAE | DINO | CLIP (token) | CLIP (keys) | SD | SD + DINO + CLIP |
|---|---|---|---|---|---|---|
| **VOC** | 54.9 | 59.1 | 51.4 | 61.8 | 64.4 | 66.4 |

**"Stuff" filtering.** Occasionally, $c_i$ might not describe a countable object category but an identifiable region in the image, *e.g.*, sky, often referred to as a "stuff" class. "Stuff" classes warrant additional consideration as they might appear as background in images of other categories, *e.g.*, boat images might often contain regions of water and sky. As a result, the process outlined above might sample background prototypes for one class that coincide with the foreground prototypes of another. To mitigate this issue, we introduce an additional filtering step to detect and reject such prototypes, when the full vocabulary, *i.e.*, the set of classes under consideration, is known. First, we only consider foreground prototypes for "stuff" classes. Additionally, any negative prototypes of "thing" classes with high cosine similarity with any of the "stuff" class prototypes are simply removed. In our experiments, we use ChatGPT [45] to automatically categorise a set of classes as "thing" or "stuff".

## 4. Experiments

We evaluate OVDiff on the open-vocabulary semantic segmentation task. First, we consider different feature extractors and investigate how they can be grounded by leveraging our approach. We then compare our method with prior work. We ablate the components of OVDiff, visualize the prototypes, and conclude with a qualitative comparison with prior works on in-the-wild images.

**Datasets and implementation details.** As the approach does not require further training of components, we only

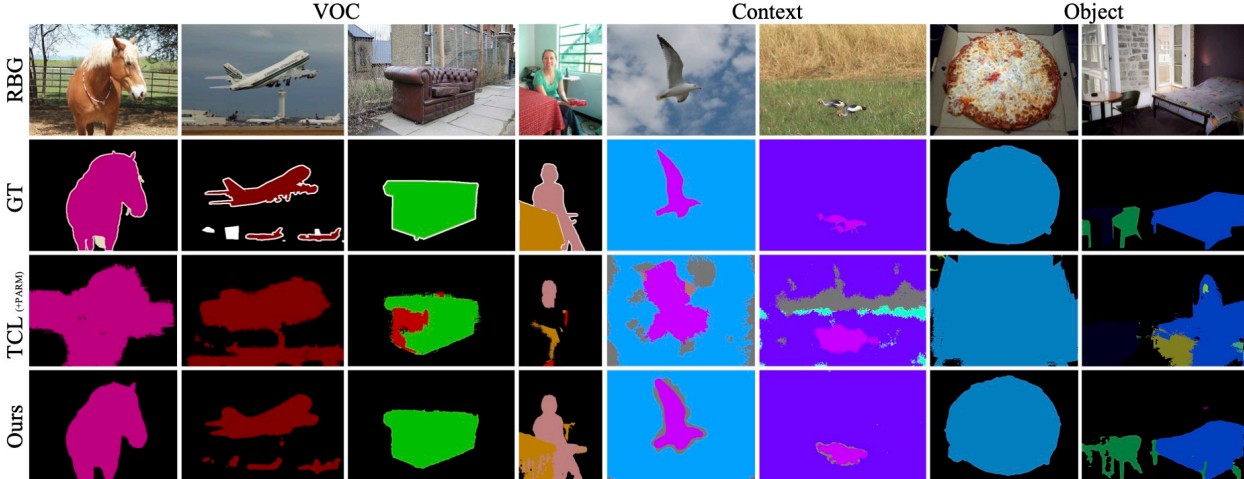

Figure 3. Qualitative results. OVDiff in comparison to TCL (+ PAMR). OVDiff provides more accurate segmentations across a range objects and stuff classes with well defined object boundaries that separate from the background well.

consider data for evaluation. Following prior work [70], to assess the segmentation performance, we report mean Intersection-over-Union (mIoU) on validation splits of PASCAL VOC (VOC) [18], PASCAL Context (Context) [42] and COCO-Object (Object) [7] datasets, with 20, 59, and 80 foreground classes, respectively. These datasets include a background class to reflect a realistic setting of non-exhaustive vocabularies. Context also contains both "things" and "stuff" classes. We also evaluate without background on VOC, Context, ADE20K [78], COCO-Stuff [7] and Cityscapes [14], with 20, 59, 150, 171, and 19 classes, respectively, but do not consider this a realistic setting as it relies on knowing which pixels cannot be described by a set of categories. Thus we leave such evaluation to Appendix A.3. Similar to [9, 70, 71], we employ a sliding window approach. We use two scales to aid with the limited resolution of off-the-shelf feature extractors with square window sizes of 448 and 336 and a stride of 224 pixels. We set the size of the support set to $N = 32$. For the diffusion model, we use Stable Diffusion v1.5; for unsupervised segmenter $\Gamma$, we employ CutLER [64].

### 4.1. Grounding feature extractors

Our method can be combined with *any* pretrained visual feature extractor for constructing prototypes and extracting image features. To verify this quantitatively, we experiment with various self-supervised ViT feature extractors (Tab. 2): DINO [8], MAE [23], and CLIP [46]. We also use SD as a feature extractor.

We find that SD performs the best, though CLIP and DINO also show strong performance based on our experiments on VOC. MAE shows the weakest performance, which may be attributed to its lack of semanticity [23]; yet it is still competitive with the majority of purposefully trained networks when employed as part of our approach. We find that taking *keys* of the second to last layer in CLIP yields better

results than using patch tokens (CLIP token). As feature extractors have different training objectives, we hypothesise that their feature spaces might be complementary. Thus, we also consider an ensemble approach. In this case, the cosine distances formed between features of different extractors and respective prototypes are averaged. The combination of SD, DINO, and CLIP performs the best. We adopt this formulation for the main set of experiments.

### 4.2. Comparison to existing methods

In Tab. 1, we compare our method with prior work that does not rely on manual mask annotation on three datasets: VOC, Context, Object. We include a brief overview of the methods in the supplement. We find that our method compares favourably, outperforming other methods in all settings. In particular, results on VOC show the largest margin, with more than 5% improvement over prior work.

We also consider a version of our method, OVDiff (-CutLER), that does not rely on an additional unsupervised segmenter $\Gamma$. Instead, the attention masks are thresholded. We observe that such a version of OVDiff has strong performance, outperforming prior work as well. CutLER is helpful, but not a critical component, and OVDiff performs strongly without it.

In the same table, we also combine our method with PAMR [1], the post-processing approach employed by TCL. We find that it improves results for our method, though improvements are less drastic since our method already yields better segmentation and boundaries.

Qualitative results are shown in Fig. 3. This figure highlights a key benefit of our approach: the ability to exploit contextual priors through the use of background prototypes, which in turn allows for the direct assignment of pixels to a background class. This improves segmentation quality because it makes it easier to differentiate objects from the

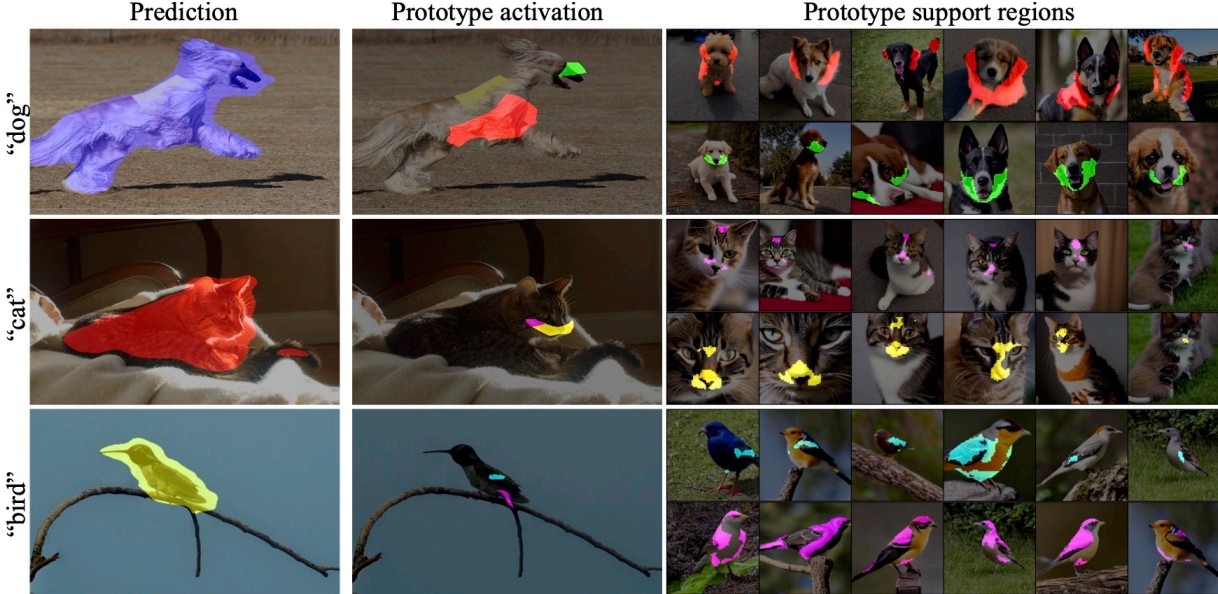

|  | Prediction | Prototype activation | Prototype support regions |
|---|---|---|---|

Figure 4. Analysis of the segmentation output by linking regions to samples in the support set. Left: our results for different classes. Middle: select color-coded regions "activated" by different prototypes for the class. Right: regions in the support set images corresponding to these (part-level) prototypes.

Table 3. Ablation of different components. Each component is removed in isolation, measuring the drop ($\Delta$) in mIoU on VOC and Context datasets. Using SD features.

| Configuration | VOC | $\Delta$ | Context | $\Delta$ |
|---|---|---|---|---|
| Full | 64.4 | | 29.4 | |
| w/o bg prototypes | 53.2 | -11.2 | 28.9 | -0.5 |
| w/o category filter | 54.4 | -10.0 | 25.2 | -4.2 |
| w/o "stuff" filter | n/a | | 26.9 | -2.5 |
| w/o CutLER | 60.4 | -4.0 | 27.6 | -1.8 |
| w/o sliding window | 62.2 | -2.2 | 28.6 | -0.8 |
| only average $\bar{P}$ | 62.5 | -1.9 | 28.4 | -1.0 |

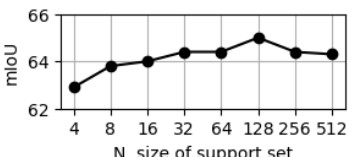

Figure 5. PascalVOC results with increasing support size $N$.

### 4.3. Ablations

Next, we ablate the components of OVDiff on VOC and Context datasets. For these experiments, only SD is employed as a feature extractor. We remove individual components and measure the change in segmentation performance, summarising the results in Tab. 3. Our first observation is that background prototypes have a major impact on performance. When removing them from consideration, we instead threshold the similarity scores of the images with the foreground prototypes (set to 0.72, determined via grid search); in this case, the performance drops significantly, which again highlights the importance of leveraging contextual priors. On Context, the impact is less significant, likely due to the fact that the dataset contains "stuff" categories. Removing the *instance-* and *part-level* prototypes also negatively affects performance. Additionally, removing the category pre-filtering has a major impact. We hypothesize that this introduces spurious correlations between prototypes of different classes. On Context, "stuff" filtering is also important. We again consider the importance of using an unsupervised segmenter, CutLER, for prototype mask extractions, using thresholding instead. We find this slightly reduces performance in this setting as well. Overall, background prototypes and pre-filtering contribute the most.

Finally, we measure the effect of varying the size of the

background and to delineate their boundaries. In comparison, TCL predictions are very coarse and contain more noise.

**Computation cost.** We focus on a construction of a method to show that existing foundational diffusion models can be used for segmentation with great efficacy without further training. OVDiff requires computing prototypes instead. With our unoptimized implementation, we measure around $110 \pm 10$s to calculate prototypes using SD for a single class, or around 1.14 TFLOP/s-hours of compute. While the focus of this study is not computational efficiency, we can compare prototype sampling to the cost of additional training of other methods: TCL requires 2688, GroupViT 10752, and OVSegmentor 624 TFLOP/s-hours.[1] While training has an upfront compute cost and requires special infrastructure (*e.g.* OVSegmentor uses 16×A100s), OVDiff's prototype set can be grown progressively as needed, while showing better performance.

---

[1]Estimated as training time × num. GPUs × theoretical peak TFLOP/s for GPU type.

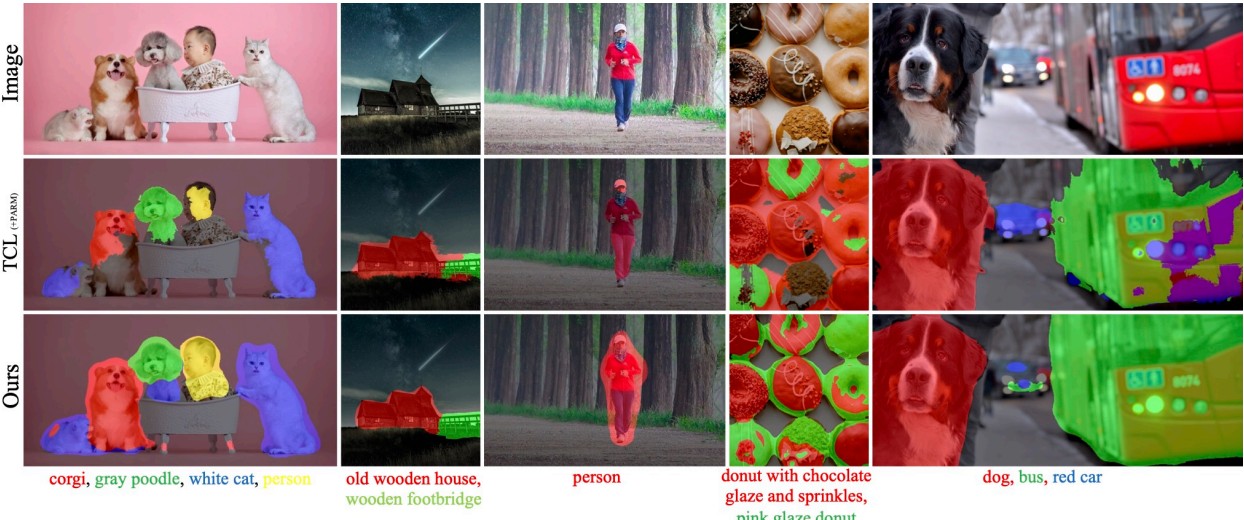

Figure 6. Qualitative comparison on challenging in-the-wild images with TCL, which struggles with object boundaries, missing parts of objects, or including surroundings. Our method has more appropriate boundaries and makes fever errors overall, but does produce a small halo effect around objects due to the upscaling of feature extractors.

support set $N$ in Fig. 5. We find that OVDiff already shows strong performance even at a low number of samples for each query. With increasing the number of samples, the performance improves, saturating at around $N = 32$. which we use in our main experiments.

### 4.4. Explaining segmentations

We inspect how our method segments certain regions by considering which prototype from $\mathcal{P}_c^{\mathrm{fg}}$ was used to assign a class $c$ to a pixel. Prototypes map to regions in the support set from where they were aggregated, *e.g.*, instances prototypes are associated with foreground masks $M_n^{\mathrm{fg}}$ and part prototypes with centroids/clusters. By following these mappings, a set of support image regions can be retrieved for each segmentation decision, providing a degree of explainability. Fig. 4 illustrates this for examples of dog, cat, and bird classes. For visualisation purposes, selected prototypes and corresponding regions are shown. On the left, we show the full segmentation result of each image. In the middle, we select regions that correlate best with certain class prototypes. On the right, we retrieve images from the support set and highlight where each prototype emerged. We find that meaningful part segmentation merges due to clustering the support image features, and similar regions are segmented by corresponding prototypes. However, sometimes region covered in the input image will not fully align with the whole prototype (*e.g.* cat's face around the eyes or lower belly/tail of bird). Each segmentation is explained by precise regions in a small support set.

### 4.5. In-the-wild

In Fig. 6, we investigate OVDiff on chal lenging in-the-wild images with simple and complex backgrounds. We compare with TCL+PAMR. In the first three images, both methods

correctly detect the objects identified by the queries. OVDiff has small false positive "corgi" patches. TCL however misses large parts of the objects, such as most of the person, and parts of animal bodies. The distinction between the house and the bridge in the second image is also better with OVD-iff. We also note that our segmentations sometimes have halos around objects. This is caused by upscaling the low-resolution feature extractor (SD in this case). The last two images contain challenging scenarios where both approaches struggle. The fourth image only contains similar objects of the same type. Both methods incorrectly identify plain donuts as either of the specified queries. OVDiff however correctly identifies chocolate donuts with varied sprinkles and separates all donuts from the background. In the final picture, the query "red car" is added, although no such object is present. The extra query causes TCL to incorrectly identify parts of the red bus as a car. Both methods incorrectly segment the gray car in the distance. However, overall, our method is more robust and delineates objects better despite the lack of specialized training or post-processing.

## 5. Conclusion

We introduce OVDiff, an open-vocabulary segmentation method that operates in two stages. First, given queries, support images are sampled and their features are extracted to create class prototypes. These prototypes are then compared to features from an inference image. This approach offers multiple advantages: diverse prototypes accommodating various visual appearances and negative prototypes for background localisation. OVDiff outperforms prior work on benchmarks, exhibiting fewer errors, effectively separating objects from background, and providing explainability through segmentation mapping to support set regions.

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
