# Supplementary Material

In this supplementary material, we provide additional experimental results, including further ablations and qualitative comparisons (Appendix A), consider the limitations and broader impacts of our work (Appendix B), and conclude with additional details concerning the implementation (Appendix C).

## A. Additional experiments

This section provides additional experimental results of OVDiff.

### A.1. Additional Comparisons

**Category filter.** To ensure that the category pre-filtering does not give our approach an unfair advantage, we augment two methods (TCL [9] and OVSegmentor [71], which are the closest baselines with code and checkpoints available) with our category pre-filtering. We evaluate on the Pascal VOC dataset (where the category filter shows a significant impact; see Table 3) and report the results in Tab. A.1. We observe that TCL improves by 0.6, while the performance of OVSegmentor drops by 0.1. On the contrary, our method benefits substantially from this component, but it still shows stronger performance without the filter than baselines with.
**Influence of $\Gamma$ segmentation method.** We also further investigate the use of CutLER [64] to obtain segmentation masks. We also provide example results of segmentation in Fig. C.4. In Tab. A.2, we devise a baseline where CutLER-predicted masks are used to average the CLIP image encoder's final spatial tokens after projection. Averaged tokens are compared with CLIP text embeddings to assign a class. While relying on pre-trained components (like ours), this avoids support set generation. In the same table, we also consider whether the objectness prior provided by CutLER could be beneficial to other methods as well. We consider a version of TCL [9] and OVSegmentor [71] which we augment with CutLER. That is, after methods assign class probabilities to each pixel/patch, a majority voting for a class is performed in every region predicted by CutLER. This combines CutLER's understanding of objects and their boundaries, aspects where prior methods struggle, with open-vocabulary segmentation. However, we observe that this negatively impacts the performance of these methods, which we attribute to only a limited performance of CutLER in complex scenes present in the datasets. Finally, we also include a version of OVDiff that does not rely on CutLER for mask extractions, instead using thresholded masks. We observe that such a version of our method also has strong performance.

We additionally experiment with stronger segmenters to understand the influence of FG/BG mask quality. We replace our FG/BG segmentation approach with strong supervised models: with SAM, we achieve 67.1 on VOC, and with

Table A.1. Use of category filter component. OVDiff without category filter outperforms prior work with cat. filter.

| Model | Category filter | |
|---|---|---|
| | ✗ | ✓ |
| OVSegmentor | 53.8 | 53.7 |
| TCL | 51.2 | 51.8 |
| TCL (+PAMR) | 55.0 | 56.0 |
| OVDiff | **56.2** | **66.4** |

Table A.2. Application of CutLER. Prior work does not benefit from using CutLER during inference, while OVDiff shows strong results without it.

| Model | CutLER | VOC | Context | Object |
|---|---|---|---|---|
| CLIP | ✓ | 33.0 | 11.6 | 11.1 |
| OVSegmentor | | 53.8 | 20.4 | 25.1 |
| OVSegmentor | ✓ | 38.7 | 14.4 | 16.8 |
| TCL | | 51.2 | 24.3 | 30.4 |
| TCL | ✓ | 43.1 | 20.5 | 22.7 |
| OVDiff | | 62.8 | 28.6 | 34.9 |
| OVDiff | ✓ | **66.3 ± 0.2** | **29.7 ± 0.3** | **34.6 ± 0.3** |

Grounded SAM, 68.5. This slightly improves results from 66.3 of our configuration with CutLER, but the performance gain is not large and thus not critical.
**Class prompts.** We additionally consider whether corrections introduced to class prompts might have similarly provided additional benefits to our approach (see Appendix C.3 for details). To that end, we also evaluate TCL and OVSegmenter (methods that do not rely on additional prompt curation) with our corrected prompts and consider a version of our method without such corrections in Tab. A.3. We observe only marginal to no impact on the performance.
**Prompt template** Finally, we consider the prompt template employed when sampling support image set: "A good picture of a $\langle c_i \rangle$" for class prompt $c_i$. This template is generic and broadly applicable to virtually any natural language specification of a target class. While prior work adopts prompt expansion by considering a list of synonyms and subcategories, it is not entirely clear how such a strategy could be systematically performed for any in-the-wild prompts, such as a "chocolate glazed donut". We experiment with a list of synonyms and subclasses, as employed by [48], on VOC datasets measuring 66.4 mIoU, which is similar to our single prompt performance $66.3 \pm 0.2$. Curating such lists automatically is an interesting future scaling direction.

### A.2. Additional ablations

**Prototype combinations.** In Tab. A.6, we consider the three different types of prototypes described in Section 3 and test their performance individually and in various combinations. We find that the "part" prototypes obtained by $K$-means

Table A.3. Using corrected prompts. We consider if corrected class names benefit prior work. We observe negligible to no effect.

| Model | Correction | VOC | Context | Object |
|---|---|---|---|---|
| OVSegmentor | | 53.8 | 20.4 | 25.1 |
| OVSegmentor | ✓ | 53.9 | 20.4 | 25.1 |
| TCL | | 51.2 | 24.3 | 30.4 |
| TCL | ✓ | 50.6 | 24.3 | 30.4 |
| OVDiff | | 66.1 | 29.5 | 34.9 |
| OVDiff | ✓ | **66.3 ± 0.2** | **29.7 ± 0.3** | **34.6 ± 0.3** |

Table A.4. Choice of $K$ for number of centroids.

| K | VOC | Context |
|---|---|---|
| 8 | 63.8 | 29.2 |
| 16 | 64.0 | 29.3 |
| 32 | 64.4 | 29.4 |
| 64 | 64.3 | 28.0 |

Table A.5. Ablation of different SD feature configurations. Removing first and last cross attention *layers*, mid, $1^{st}$ and $2^{nd}$ upsampling *blocks* (all layers in the block) has a negative effect.

| 1st layer | Mid block | Up-1 block | Up-2 block | Last layer | Context |
|---|---|---|---|---|---|
| ✓ | ✓ | ✓ | ✓ | ✓ | 29.4 |
| | ✓ | ✓ | ✓ | ✓ | 29.4 |
| ✓ | | ✓ | ✓ | ✓ | 29.2 |
| ✓ | ✓ | | ✓ | ✓ | 27.3 |
| ✓ | ✓ | ✓ | | ✓ | 28.9 |
| ✓ | ✓ | ✓ | ✓ | | 29.3 |

Table A.6. Ablation of various configurations for prototypes. We consider average $\bar{P}$, instance $P_n$, and part $P_k$ prototypes individually and in various combinations on VOC and Context datasets. Combination of all three types of prototypes shows strongest results.

| $\bar{P}$ | $P_n$ | $P_k$ | VOC | Context |
|---|---|---|---|---|
| ✓ | ✓ | ✓ | 64.4 | 29.4 |
| ✓ | | ✓ | 61.7 | 29.3 |
| ✓ | ✓ | | 63.5 | 29.4 |
| | ✓ | ✓ | 62.5 | 28.4 |
| | | ✓ | 63.7 | 28.8 |
| | ✓ | | 60.0 | 29.0 |
| ✓ | | | 62.5 | 28.4 |

clustering show strong performance when considered individually on VOC. Instance prototypes show strong individual performance on Context, as well as in combination with the average category prototype. The combination of all three types shows the strongest results across the two datasets, which is what we adopt in our main set of experiments.

We also consider the treatment of prototypes under the

stuff filter. We investigate the impact of not excluding background prototypes for "stuff" classes. In this setting, we measure 29.1 on Context, which is a slight reduction in performance. We also investigate the benefit of categorisation into "things" and "stuff" used in the stuff filter component. Instead, we filter all background prototypes using all foreground prototypes. In this configuration, we measure 27.6 on Context. Both configurations show a reduction from 29.4, measuring using the stuff filter with categorisation in "stuff" and "things", as used in our main experiments. Finally, we experiment by removing part-level prototypes for "stuff" classes, which also results in a performance drop to 28.0.

$K$ **- number of clusters.** In Tab. A.4, we investigate the sensitivity of the method to the choice of $K$ for the number of "part" prototypes extracted using $K$-means clustering. Although our setting $K = 32$ obtains slightly better results on Context and VOC, other values result in comparable segmentation performance suggesting that OVDiff is not sensitive to the choice of $K$ and a range of values is viable.

**SD features.** When using Stable Diffusion as a feature extractor, we consider various combinations of layers/blocks in the UNet architecture. We follow the nomenclature used in the Stable Diffusion implementation where consecutive layers of Unet are organised into *blocks*. There are 3 downsampling blocks with 2 cross-attention layers each, a mid-block with a single cross-attention, and 3 up-sampling blocks with 3 cross-attention layers each. We report our findings in Tab. A.5. Including the first and last cross-attention layers in the feature extraction process has a small positive impact on segmentation performance, which we attribute to the high feature resolution. We also consider excluding features from the middle block of the network due to small $8 \times 8$ resolution but observe a small negative impact on performance on the Context dataset. We also investigate whether including the first (Up-1) and the second upsampling (Up-2) blocks are necessary. Without them, the performance drops the most out of the configurations considered. Thus, we use a concatenation of features from the middle, first and second upsampling blocks and the first and last layers in our main experiments.

## A.3. Evaluation without background

One of the notable advantages of our approach is the ability to represent background regions via (negative) prototypes, leading to improved segmentation performance. Nevertheless, we hereby also evaluate our method under a different evaluation protocol adopted in prior work, which excludes the *background* class from the evaluation. We note that prior work often requires additional considerations to handle background, such as thresholding. In this setting, however, the background class is *not* predicted, and the set of categories, thus, must be exhaustive. As in practice, this is not the case, and datasets contain unlabelled pixels

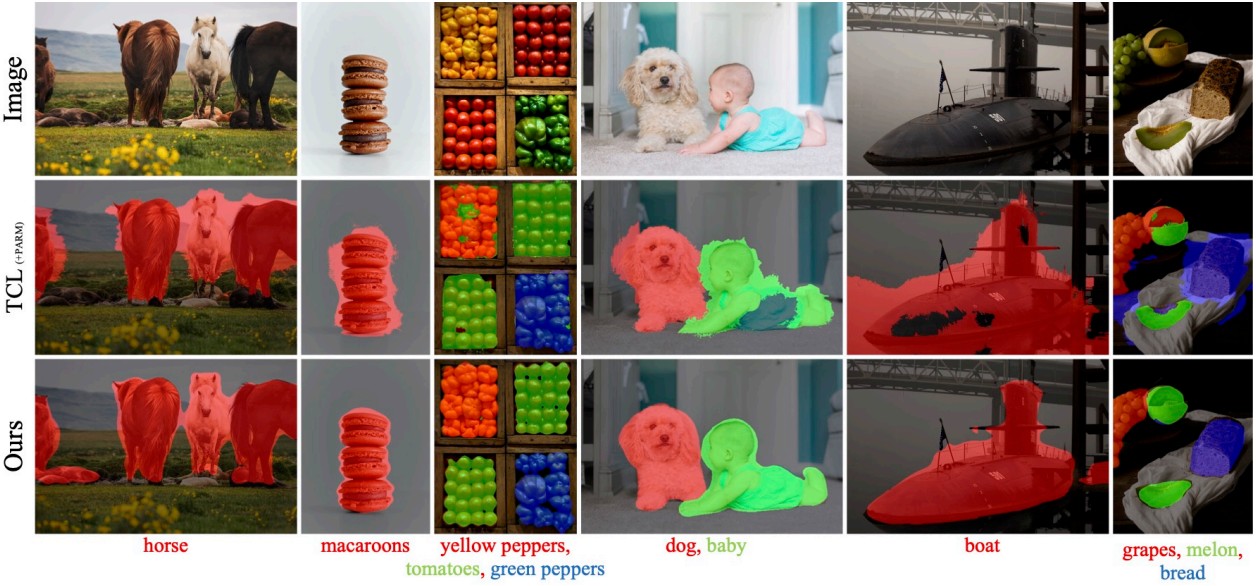

Figure A.1. Qualitative comparison on in-the-wild images. OVDiff performs significantly better than prior state-of-the-art, TCL, on wildlife images containing multiple instances, studio photos with simple backgrounds, images containing multiple categories and an image containing a rare instance of a class.

Table A.7. Comparison with methods when background is excluded (decided by ground truth). OVDiff shows comparable performance to prior works despite only relying on pretrained feature extractors. * result from [9].

| Method | VOC-20 | Context-59 | ADE | Stuff | City |
|---|---|---|---|---|---|
| CLIPpy | – | – | 13.5 | – | – |
| OVSegmentor | – | – | 5.6 | – | – |
| GroupViT* | 79.7 | 23.4 | 9.2 | 15.3 | 11.1 |
| MaskCLIP* | 74.9 | 26.4 | 9.8 | 16.4 | 12.6 |
| ReCo* | 57.5 | 22.3 | 11.2 | 14.8 | 21.1 |
| TCL | 77.5 | 30.3 | **14.9** | 19.6 | 23.1 |
| **OVDiff** | **80.9** | **32.9** | 14.1 | **20.3** | **23.4** |

(or simply a background label), such image areas are removed from consideration. Consequently, less emphasis is placed on object boundaries in this setting. As in this setting the background prediction is invalid, we do not consider negative prototypes. For this setting, we benchmark on 5 datasets following [9]: PascalVOC without background, termed VOC-20, Pascal Context without background, termed Context-59, and ADE20k [78], which contains 150 foreground classes, termed ADE-150, COCO-Stuff, termed Stuff, and Cityscapes, termed City. This setting tests the ability of various methods to discriminate between different classes, which for OVDiff is inherent to the choice of feature extractors. Despite this, our method shows competitive performance accross wide range of benchmarks Tab. A.7.

## A.4. Qualitative results

We include additional qualitative results from the benchmark datasets in Fig. A.2. In Fig. A.3, we show examples of support images sampled for some things, and stuff categories. In Fig. C.5, we show examples of support set images sampled for rare *pikachu* class.

## B. Broader impact

Semantic segmentation is a component in a vast and diverse spectrum of applications in healthcare, image processing, computer graphics, surveillance and more. As for any foundational technology, applications can be good or bad. OVDiff is similarly widely applicable. It also makes it easier to use semantic segmentation in new applications by leveraging existing and new pre-trained models. This is a bonus for inclusivity, affordability, and, potentially, environmental impact (as it requires no additional training, which is usually computationally intensive); however, these features also mean that it is easier for bad actors to use the technology.

Because OVDiff does not require further training, it is more versatile but also inherits the weaknesses of the components it is built on. For example, it might contain the biases (e.g., gender bias) of its components, in particular Stable Diffusion [53], which is used for generating support images for any given category/description. Thus, it should not be exposed without further filtering and detection of, e.g., NSFW material in the sampled support set. Finally, OVDiff is also bound by the licenses of its components.

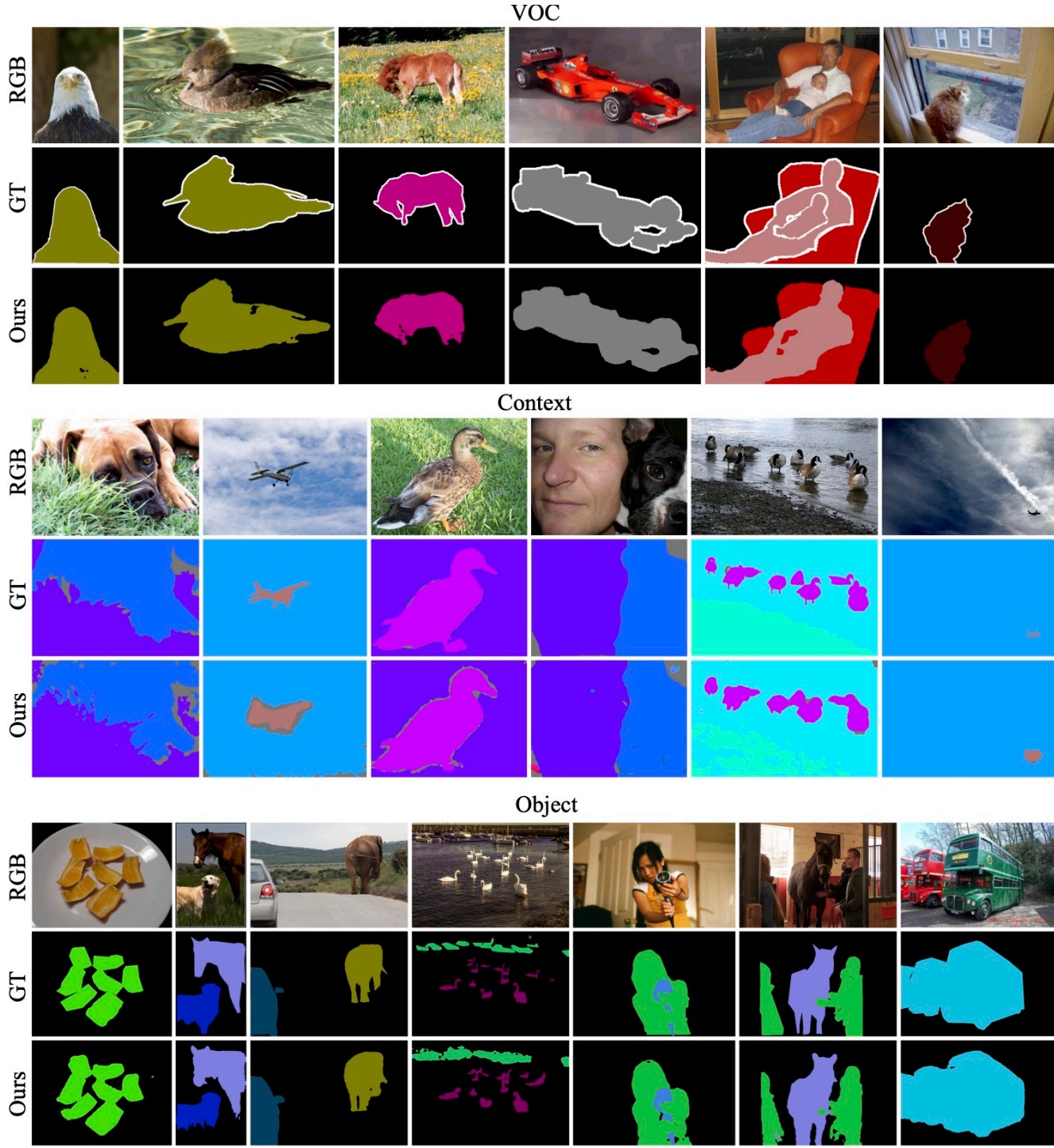

Figure A.2. Additional qualitative results. Images from Pascal VOC (top), Pascal Context (middle), and COCO Object (bottom).

## B.1. Limitations

As OVDiff relies on pretrained components, it inherits some of their limitations. OVDiff works with the limited resolution of feature extractors, due to which it might occasionally miss tiny objects. Furthermore, OVDiff cannot segment what the generator cannot generate. For example, current diffusion models struggle with producing legible text, which can make it difficult to segment specific words. Furthermore, applications in domains far from the generator's training data

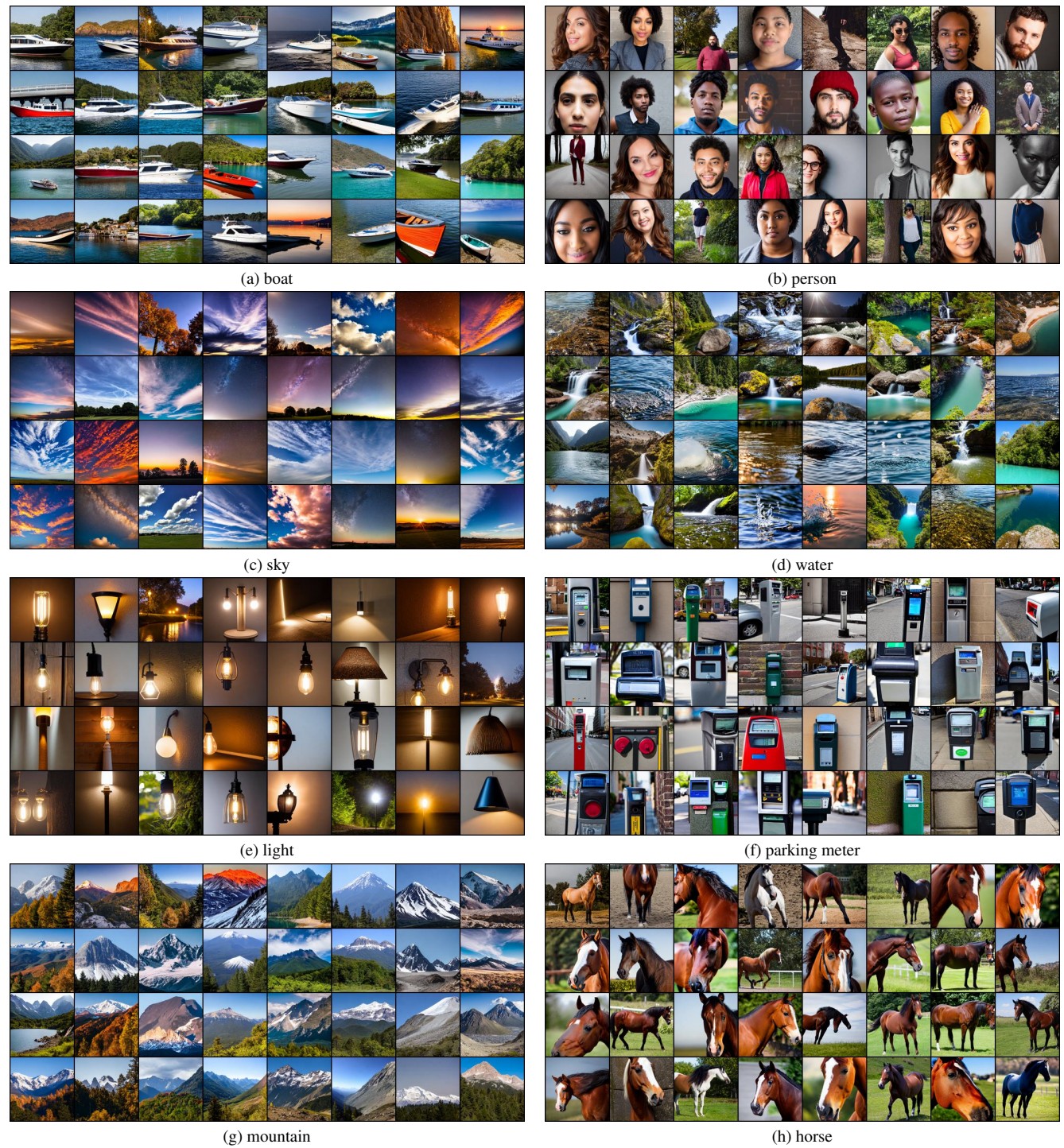

(a) boat                  (b) person

(c) sky                  (d) water

(e) light                 (f) parking meter

(g) mountain             (h) horse

Figure A.3. Images sampled for a support set of some categories.

(*e.g.* medical imaging) are unlikely to work out of the box.

## C. OVDiff: Further details

In this section, we provide additional details concerning the implementation of OVDiff. We begin with a brief overview of the attention mechanism and diffusion models central to

extracting features and sampling images. We review different feature extractors used. We specify the hyperparameter setting for all our experiments and provide an overview of the exchange with ChatGPT used to categorise classes into "thing" and "stuff".

## C.1. Preliminaries

**Attention.** In this work, we make use of pre-trained ViT [16] networks as feature extractors, which repeatedly apply multi-headed attention layers. In an attention layer, input sequences $X \in \mathbb{R}^{l_x \times d}$ and $Y \in \mathbb{R}^{l_y \times d}$ are linearly project to forms *keys*, *queries*, and *values*: $K = W_k Y$, $Q = W_q X$, $V = W_v X$. In self-attention, $X = Y$. Attention is calculated as $A = \text{softmax}(\frac{1}{\sqrt{d}} Q K^\top)$, and softmax is applied along the sequence dimension $l_y$. The layer outputs an update $Z = X + A \cdot V$. ViTs use multiple heads, replicating the above process in parallel with different projection matrices $W_k, W_q, W_v$. In this work, we consider *queries* and *keys* of attention layers as points where useful features that form meaningful inner products can be extracted. As we detail later (Appendix C.2), we use the *keys* from attention layers of ViT feature extractors (DINO/MAE/CLIP), concatenating multiple heads if present.

**Text-to-image diffusion models.** Diffusion models are a class of generative models that form samples starting with noise and gradually denoising it. We focus on latent diffusion models [50] which operate in the latent space of an image VAE [28] forming powerful conditional image generators. During training, an image is encoded into VAE latent space, forming a latent vector $z_0$. A noise is injected forming a sample $z_\tau \sim \mathcal{N}(z_\tau; \sqrt{1 - \alpha_\tau} z_0, \alpha_\tau I)$ for timestep $\tau \in \{1 \dots T\}$, where $\alpha_\tau$ are variance values that define a noise schedule such that the resulting $z_T$ is approximately unit normal. A conditional UNet [51], $\epsilon_\theta(z_t, t, c)$, is trained to predict the injected noise, minimising the mean squared error $\mathbb{E}_t \left( \alpha_t \| \epsilon_\theta(z_t, t, c) - z_0 \|_2 \right)$ for some caption $c$ and additional constants $a_t$. The network forms new samples by reversing the noise-injecting chain. Starting from $\hat{z}_T \sim \mathcal{N}(\hat{z}_T; 0, I)$, one iterates $\hat{z}_{t-1} = \frac{1}{\sqrt{1 - \alpha_t}} (\hat{z}_t + \alpha_t \epsilon_\theta(\hat{z}_t, t, c)) + \sqrt{\alpha_t} \hat{z}_t$ until $\hat{z}_0$ is formed and decoded into image space using the VAE decoder. The conditional UNet uses cross-attention layers between image patches and language (CLIP) embeddings to condition on text $c$ and achieve text-to-image generation.

## C.2. Feature extractors

OVDiff is buildable on top of any pre-trained feature extractor. In our experiments, we have considered several networks as feature extractors with various self-supervised training regimes:

- **DINO** [8] is a self-supervised method that trains networks by exploring alignment between multiple views using an exponential moving average teacher network. We use the ViT-B/8 model pre-trained on ImageNet[2] and extract features from the *keys* of the last attention layer.

- **MAE** [22] is a self-supervised method that uses masked image inpainting as a learning objective, where a portion of image patches are dropped, and the network seeks to reconstruct the full input. We use the ViT-L/16 model pre-trained on ImageNet at a resolution of 448 [27].[3] The *keys* of the last layer of the *encoder* network are used. No masking is performed.

- **CLIP** [46] is trained using image-text pairs on an internal dataset WIT-400M. We use ViT-B/16 model[4]. We consider two locations to obtain dense features: *keys* from a self-attention layer of the image encoder and *tokens* which are the outputs of transformer layers. We find that *keys* of the second-to-last layer give better performance.

- We also consider **Stable Diffusion**[5] (v1.5) itself as a feature extractor. To that end, we use the *queries* from the cross-attention layers in the UNet denoiser, which correspond to the image modality. Its UNet is organised into three downsampling blocks, a middle block, and three upsampling blocks. We observe that the middle layers have the most semantic content, so we consider the middle block, 1st and 2nd upsampling blocks and aggregate features from all three cross-attention layers in each block. As the features are quite low in resolution, we include the first downsampling cross-attention layer and the last upsampling cross-attention layer as well. The feature maps are bilinearly upsampled to resolution $64 \times 64$ and concatenated. A noise appropriate for $\tau = 200$ timesteps is added to the input. For feature extraction, we run SD in *unconditional* mode, supplying an empty string for text caption.

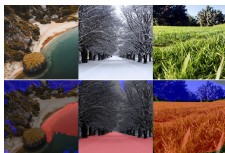 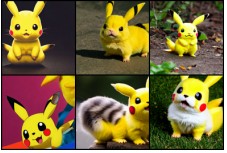

Figure C.4. FG/BG segmentation of classes of *water*, *snow* and *grass*. The foreground is in red, while the background is shown in blue.

Figure C.5. Example images from the support set of a rare *pikachu* class.

---

[2]Model and code available at https://github.com/facebookresearch/dino.
[3]Model and code from https://github.com/facebookresearch/long_seq_mae.
[4]Model and code from https://github.com/openai/CLIP.
[5]We use implementation from https://github.com/huggingface/diffusers.

## C.3. Datasets

We evaluate on validation splits of PASCAL VOC (VOC), Pascal Context (Context) and COCO-Object (Object) datasets. PASCAL VOC [17, 18] has 21 classes: 20 foreground plus a background class. For Pascal Context [42], we use the common variant with 59 foreground classes and 1 background class. It contains both "things" and "stuff" classes. The COCO-Object is a variant of COCO-Stuff [7] with 80 "thing" classes and one class for the background. Textual class names are used as natural language specifications of names. We renamed or specified certain class names to fix errors (*e.g.* `pottedplant` → `potted plant`), resolve ambiguity better (*e.g.* `mouse` → `computer mouse`) or change to more common spelling/word (*e.g.* `aeroplane` → `airplane`), resulting in 14 fixes. We experiment and measure the impact of this in Appendix A.1 for our and prior work.

## C.4. Comparative baselines

We briefly review the prior work in used in our experiments, mainly in Table 1. We consider baselines that do not rely on mask annotations and have code and checkpoints available or detail their evaluation protocol that matches that used in other prior works [9, 70, 71].Most prior work [9, 35, 37, 49, 70, 71] trains image and text encoders on large image-text datasets with a contrastive loss. The methods mainly differ in their architecture and use of grouping mechanisms to ground image-level text on regions. ViL-Seg [35] uses online clustering, GroupViT [70] and ViewCo [49] employ group tokens. OVSegmentor [71] uses slot-attention and SegCLIP [37] a grouping mechanism with learnable centers. CLIPPy [48], TCL [9], and MaskCLIP [79] predict classes for each image patch: [48] use max-pooling aggregation, [9] self-masking, and [79] modify CLIP for dense predictions. To assign a background label [9, 35, 37, 49, 70] use thresholding while [48] uses dataset-specific prompts. CLIP-DIY [68] leverages CLIP as a zero-shot classifier and applies it on multiple scales to form a dense segmentation. ReCO [56] is closer in spirit to our approach as it uses a support set for each prompt; this set, however, is CLIP-retrieved from curated image collections, which may not be applicable for any category in-the-wild.

We also note that prior work builds on top of similar pre-trained components such as CLIP in [9, 37, 56, 79], OpenCLIP in [68], DINO + T5/RoBERTa in [48, 71]. We additionally make use of StableDiffusion, which is trained on a larger dataset (3B, compared to 400M of CLIP or 2B or OpenCLIP). OVDiff is, however, fundamentally different to all prior work, as (a) it generates a support set of synthetic images given a class description, and (b) it does not rely on additional training data and further training for learning to segment.

## C.5. Hyperparameters

OVDiff has relatively few hyperparameters and we use the same set in all experiments. Unless otherwise specified, $N = 32$ images are sampled using classifier-free guidance scale [25] of 8.0 and 30 denoising steps. We employ `DPM-Solver` scheduler [36]. When sampling images for the support sets, we also use a negative prompt "*text, low quality, blurry, cartoon, meme, low resolution, bad, poor, faded*". If/when segmenter $\Gamma$ fails to extract any components in a sampled image, a fallback of adaptive thresholding of $A_n$ is used, following [34]. During inference, we set $\eta = 10$, which results in 1024 text prompts processed in parallel, a choice made mainly due to computational constraints. We set the thresholds for the "stuff" filter between background prototypes for "things" classes and the foreground of "stuff" at 0.85 for all feature extractors. When sampling, a seed is set for each category individually to aid reproducibility. With our unoptimized implementation, we measure around $110\pm10$s to calculate prototypes (sample images, extract features and aggregate) for a single category or $50.2\pm2$s without clustering using SD. Using CLIP, we measure $49.2 \pm 0.2$s with clustering and $47.7 \pm 0.2$s without. We note that sampling time grows linearly: we measure 55s for 16, 110s for 32, and 213s for 64 images per class. The prototype storage requirements are 0.39MB using CLIP/DINO for each class.

We additionally measure the speed of inference at 0.6s per image, which is slightly slower but comparable to 0.2s for TCL and 0.08s for OVSegmentor. We performed inference measurements using SD on the same machine with a 2080Ti GPU using 21 classes and the same resolution/sliding window settings for all methods.

## C.6. Interaction with ChatGPT

We interact with ChatGPT to categorise classes into "stuff" and "things" for the stuff filter component. Due to input limits, the categories are processed in blocks. Specifically, we input "*In semantic segmentation, there are "stuff" or "thing" classes. Please indicate whether the following class prompts should be considered "stuff" or "things":*". We show the output in Tab. C.8. Note there are several errors in the response, *e.g.* `glass`, `blanket`, and `trade name` are actually instances of tableware, bedding and signage, respectively, so should more appropriately be treated as "things". Similarly, `land` and `sand` might be more appropriately handled as "stuff", same as `snow` and `ground`. Despite this, We find ChatGPT contains sufficient knowledge when prompted with "in semantic segmentation". We have estimated the accuracy of ChatGPT in thing/stuff classification using the categories of COCO-Stuff, which are defined as 80 "things" and 91 "stuff" categories. ChatGPT achieves an accuracy rate of 88.9% in this case. We also measure the impact the potential errors have on our performance by providing "oracle" answers on the Context dataset. We measure 29.6 mIoU, which

Table C.8. **Response from interaction with ChatGPT.** We used ChatGPT model to automatically categorise classes in "stuff" or "things".

| | | | | | |
|---|---|---|---|---|---|
| airplane: | thing | window: | thing | awning: | thing |
| bag: | thing | wood: | stuff | streetlight: | thing |
| bed: | thing | windowpane: | thing | booth: | thing |
| bedclothes: | stuff | earth: | thing | television receiver: | thing |
| bench: | thing | painting: | thing | dirt track: | thing |
| bicycle: | thing | shelf: | thing | apparel: | thing |
| bird: | thing | house: | thing | pole: | thing |
| boat: | thing | sea: | thing | land: | thing |
| book: | thing | mirror: | thing | bannister: | thing |
| bottle: | thing | rug: | thing | escalator: | thing |
| building: | thing | field: | thing | ottoman: | thing |
| bus: | thing | armchair: | thing | buffet: | thing |
| cabinet: | thing | seat: | thing | poster: | thing |
| car: | thing | desk: | thing | stage: | thing |
| cat: | thing | wardrobe: | thing | van: | thing |
| ceiling: | stuff | lamp: | thing | ship: | thing |
| chair: | thing | bathtub: | thing | fountain: | thing |
| cloth: | stuff | railing: | thing | conveyer belt: | thing |
| computer: | thing | cushion: | thing | canopy: | thing |
| cow: | thing | base: | thing | washer: | thing |
| cup: | thing | box: | thing | plaything: | thing |
| curtain: | stuff | column: | thing | swimming pool: | thing |
| dog: | thing | signboard: | thing | stool: | thing |
| door: | thing | chest of drawers: | thing | barrel: | thing |
| fence: | stuff | counter: | thing | basket: | thing |
| floor: | stuff | sand: | thing | waterfall: | thing |
| flower: | thing | sink: | thing | tent: | thing |
| food: | thing | skyscraper: | thing | minibike: | thing |
| grass: | stuff | fireplace: | thing | cradle: | thing |
| ground: | stuff | refrigerator: | thing | oven: | thing |
| horse: | thing | grandstand: | thing | ball: | thing |
| keyboard: | thing | path: | thing | step: | stuff |
| light: | thing | stairs: | thing | tank: | thing |
| motorbike: | thing | runway: | thing | trade name: | stuff |
| mountain: | stuff | case: | thing | microwave: | thing |
| mouse: | thing | pool table: | thing | pot: | thing |
| person: | thing | pillow: | thing | animal: | thing |
| plate: | thing | screen door: | thing | lake: | stuff |
| platform: | stuff | stairway: | thing | dishwasher: | thing |
| plant: | thing | river: | thing | screen: | thing |
| road: | stuff | bridge: | thing | blanket: | stuff |
| rock: | stuff | bookcase: | thing | sculpture: | thing |
| sheep: | thing | blind: | thing | hood: | thing |
| shelves: | thing | coffee table: | thing | sconce: | thing |
| sidewalk: | stuff | toilet: | thing | vase: | thing |
| sign: | thing | hill: | thing | traffic light: | thing |
| sky: | stuff | countertop: | thing | tray: | stuff |
| snow: | stuff | stove: | thing | ashcan: | thing |
| sofa: | thing | palm: | thing | fan: | thing |
| table: | thing | kitchen island: | thing | pier: | thing |
| track: | stuff | swivel chair: | thing | crt screen: | thing |
| train: | thing | bar: | thing | bulletin board: | thing |
| tree: | thing | arcade machine: | thing | shower: | thing |
| truck: | thing | hovel: | thing | radiator: | thing |
| monitor: | thing | towel: | thing | glass: | stuff |
| wall: | stuff | tower: | thing | clock: | thing |
| water: | stuff | chandelier: | thing | flag: | thing |

is similar to $29.7\pm0.3$ of using ChatGPT, showing that small errors do not drastically affect the method, however, enable using "stuff" filter component, which improves performance (see Table 3).