# OpenReview forum: "Diffusion Models for Open-Vocabulary Segmentation"
_thecvf.com/CVPR/2024/Workshop/SyntaGen — SyntaGen 2024_

### Official Review · Reviewer_XcQn · 2024-03-30
**The idea is interesting, and the results support the authors' claim. Thus, I vote for acceptance.**

**Rating:** 8
**Confidence:** 4

**Review:**

## Summary

This paper proposes a training-free open-vocabulary segmentation method called OVDiff. It segments an image in three steps: generation, representation, and matching. Given a text prompt, OVDiff first generates a support set of images using a text-to-image diffusion model. Then, it uses a feature extractor to extract the features that represent the category. Finally, it uses nearest-neighbor matching to segment the target image.

## Strengths

1. This is an interesting approach that uses synthetic images of a specific category to construct a segmentor of this class.
2. OVDiff shows competitive results over baselines.
3. The paper is well-written, and the experiments are thorough and complete.

## Weaknesses

1. This method requires a long time to construct a segmentor for a class, making it less practical compared to feed-forward open-vocabulary segmentation methods.
2. OVDiff requires the generated images to have clear background-foreground separation, while this is not guaranteed for any text-to-image models.
3. While it is an interesting idea to use synthetic data as a support set, it also brings the biases of the generation models. If the query images have a significant gap with the generation model, OVDiff might fail.

---

### Official Review · Reviewer_kFSL · 2024-04-01
**Good method with extensive experiments.**

**Rating:** 9
**Confidence:** 4

**Review:**

The paper presents an open-vocabulary method that leverages foundation models instead of training a new model and requires no training or annotations. The experiment section is comprehensive with many interesting insights. The results are reasonable for a no-training method.

### Pros
- A pipeline that uses a text-to-image generative model to create visual prototypes for each class. I think this idea is innovative and should work very well. Most open-vocab methods compare visual features with text features directly, which is proven to have some domain gap even in the same latent space.
- No training, no annotation method. The pipeline works well with a surprisingly low number of support images.
- Extensive experiments and analysis, including alternative feature extractor and computation analysis.

### Cons
- I think with how advanced open-source segmentation models are, the results' segmentation masks could have had better precision.

---

### Official Review · Reviewer_BJ2L · 2024-04-04

**Rating:** 6
**Confidence:** 4

**Review:**

Summary: This paper proposes OVDiff, a novel open-vocabulary segmentation method that leverages text-to-image diffusion models SD 1.5 to generate support image sets. Using those generated support image sets to generate prototypes for the target classes and calculate the cosine similarity to segment the object in new images. The proposed method shows good results on PASCAL VOC with a large margin, and minor improvement on many other datasets (e.g PASCAL Context, COCO-Object,...)

Pros:

- The proposed method is novel and achieves state-of-the-art (SOTA) performance on many benchmarks.
- The concept of utilizing Stable Diffusion to generate a support image set and prototype for segmentation is interesting and may apply to other downstream tasks.

Cons:

- I have a concern regarding the quality of unsupervised instance segmentation (CutLER). Additional image examples for this could benefit the reader in understanding how it works. Moreover, we could rely on the SD model to generate a foreground mask of an object similar to Dataset Diffusion: Diffusion-based Synthetic Dataset Generation for Pixel-Level Semantic Segmentation by Quang Ho Nguyen et al. (https://arxiv.org/abs/2309.14303).

- The paper uses the combination of SD + Dino + Clip to extract feature prototypes but does not show any reason why SD performs the best among these feature extractors. Moreover, the performance of other SD versions, such as SD 2.1 and SD XL, is not evaluated.

- Because this method generates support image sets from a generative model, I am concerned about the prompt selection to sample images. This method does not experiment with the effect of the prompt on the performance of the method.

- This paper lacks a comparison between synthetic data and real data. How would the performance of this method be if we could use images from a training dataset? For example, using real images from the dataset or crawling from the internet as support image sets. What is the gap between synthetic data and real data?

---

### Decision · Program_Chairs · 2024-04-06

**Decision:**

Accept

**Comment:**

This paper introduces a novel method for open-vocabulary semantic segmentation, leveraging Stable Diffusion to produce synthetic images representing specific classes for the generation of class prototypes. It has received relatively high scores (6,8,9 out of 10), underscoring its intriguing concept, strong performance, and clear writing. Nonetheless, the suggestions for an ablation study suggested by Reviewer BJ2L deserve careful consideration in the camera-ready version. Integrating these comments is expected to enhance the comprehensiveness of the paper. The final decision is to accept!